# Measuring political radicalism and extremism in surveys: Three new scales

**Sebastian Jungkunz**[1,2]*, **Marc Helbling**[3], **Nina Osenbrügge**[3]

**1** Institute of Political Science and Sociology, University of Bonn, Bonn, Germany, **2** Institute of Political Science, University of Bamberg, Bamberg, Germany, **3** Chair of Sociology, Migration and Integration, University of Mannheim, Mannheim, Germany

* sebastian.jungkunz@uni-bonn.de

## Abstract

This paper introduces three new scales to measure left- and right-wing radical as well as general extremist attitudes that can be applied across Western European countries. We therefore propose a thorough conceptualization of extremist attitudes that consists of two dimensions: general extremism, by which we understand attitudes that oppose the constitutional democratic state, and another dimension that differentiates between right- and left-wing radicalism by which we understand people who take far-reaching but often one-sided positions on political issues (e.g., on nationalism or anti-imperialism) by advocating fundamental socio-political change. Based on data from Germany, Great Britain, and the Netherlands (n = 6,201) we created short indices for general extremism and left- and right-wing radicalism. We check for convergence validity by assessing the psychometric properties of the extracted indices, i.e. their internal coherence and the degree to which a scale is able to distinguish strongly extremist and non-extremist individuals. Finally, we correlate the scales with various constructs that are likely related to extremist attitudes in order to assure external or construct validity. The results indicate that the three scales are highly valid and applicable across three Western European countries. Overall, we find that about two to four percent of citizens in each country hold left-wing or right-wing extremist attitudes.

## Introduction

This paper introduces three new scales to measure left- and right-wing radical as well as general extremist attitudes that can be applied across Western European countries. In the light of the widespread successes of populist parties in Europe many achievements of liberal democracies have been put in question by new political parties. There are, however, also organizations and parties as well as fractions within populist parties that can be defined as extremist, as they reject the principles of democracy as a whole. In Germany, for example, certain fractions of the Alternative for Germany (AfD) that has been considered a populist party for a long time [1, 2], are suspected to be extremist by the German Federal Office for the Protection of the Constitution [3]. Irrespective of their profile, it can be assumed that populist parties at the far left or far right of the political spectrum are also supported by people who are extremist [4].

**Data Availability Statement:** Replication materials are available in from the Harvard Dataverse at doi:10.7910/DVN/1W4JLK (https://doi.org/10.7910/DVN/1W4JLK).

**Funding:** Deutsche Forschungsgemeinschaft (DFG) - Project number 438614532. The funders had no role in study design, data collection and analysis, decision to publish, or preparation of the manuscript.

**Competing interests:** The authors have declared that no competing interests exist.

Several scales have already been proposed (mostly in Germany) to measure right- and left-wing extremism [5–9]. There is, however, hardly any discussion of the validity of these scales and there is therefore no standard scale or agreement on how to use them. It often remains unclear how they have been constructed and how they relate to each other. It is also unclear whether these scales are limited in their capacity to capture respondents at the extreme ends of scales, something we observe e.g. in the related concept of populist attitudes (see [10]). In many cases it seems that they measure right- or left-wing radicalism rather than extremism. Some of these scales focus for example on racism or anti-capitalism. Such attitudes might be in conflict with liberal democratic norms. People holding these views do, however, not necessarily put in question democracy as such. Furthermore, there are related concepts like right- (and left-)wing authoritarianism [11–13], authoritarian personality [14, 15], or social dominance orientation [16] that have been developed over time, but where it is unclear how they are related to political extremism.

We therefore propose a more thorough conceptualization of extremist attitudes that consists of two dimensions: The first dimension measures general extremism by which we understand attitudes that oppose the constitutional democratic state [17]. We assume that extremists on the right and the left share a similar social background and their ideologies have mutual characteristics that also point toward common enemies [9] and therefore expect to find a core "extremist belief system" which is prevalent in all forms of extremism. The second dimension then differentiates between right- and left-wing radicalism by which we understand people who take far-reaching but often one-sided positions on political issues (e.g., on nationalism or anti-imperialism) by advocating fundamental socio-political change [9, 18, 19]. The combination of these two dimensions allows us to differentiate between different extremists.

Besides a new conceptualization of extremist attitudes the aim of this paper is also to provide valid indices. In a first step, we gathered the existing items for right- and left-wing radicalism as well as general extremism in the literature. Through several rounds of surveys and validity tests in Germany, the Netherlands and Great Britain we created short indices. In a second step, we check upon convergence validity by assessing the psychometric properties of the extracted indices, i.e. their internal coherence and the degree to which a scale is able to distinguish strongly extremist and non-extremist individuals. This is especially relevant for extremist attitudes, as respondents often hide their true intentions and rate themselves in the center of such scales, which makes it hard to identify the true share of extremists [4, 5, 20]. In a final step, we correlate the scales with various constructs that are said to be related to extremist attitudes in order to assure external or construct validity. Among others, we expect a relationship between extremist attitudes and vote choice for far left- and right-wing parties.

The results indicate that the three scales are highly valid and applicable across three Western European countries. This development is crucial as previous measures for assessing attitudes towards political radicalism and extremism were either lacking validation or subject to measurement errors. The newly developed scales allow for the identification of latent radical and extremist attitudes in individuals before openly supporting radical parties, enabling the investigation of the radicalization process at an earlier stage (see further [21]). This research represents helps us to advance our understanding of the causes and consequences of radical and extremist attitudes.

## Definition and conceptualization

Extremism subsumes all attitudes, behavior, organizations and goals that oppose the constitutional democratic state [17, 22]. In case it negates the fundamental equality of human beings, it is called right-wing extremism, and if it extends the principle of equality to a degree where it

superimposes individual freedom, then it is called left-wing extremism [9]. The main element is therefore an anti-democratic posture and a disposition to consider cleavages and ambivalence within society and politics as illegitimate [17, 23, 24]. Extremist ideologies build upon a claim for absolute truth, the construction of friend-and-foe images, dogmatism, a holistic and deterministic conception of history, an identitarian construction of society, dualistic rigorism and the fundamental condemnation of the present state of being [17, 22].

Right-wing extremism (RWE) is further characterized as an ideology of inegality, which builds upon an affinity for authoritarian regimes, ethnic nationalism, xenophobia or ethnocentrism, racism, social Darwinism and anti-Semitism [25, 26]. The lack of research on left-wing extremism (LWE) has thus far provided less theoretical work on a precise definition. Common to all facets of LWE is a pursuit for a socially homogeneous community and any actions for reaching this goal are directed against the democratic constitutional state. Ideologically, LWE subsumes two streams: Marxism, which stresses the importance of state authority in the process from socialism to communism, and anarchism, which fundamentally opposes the idea of the state and political authority per se [9]. Such dichotomy makes a global definition even more problematic. Nevertheless, LWE manifests in the rejection of capitalism, globalization, militarism and imperialism, and endorses anti-Americanism, anti-fascism, anti-racism but also anti-Zionism, autonomism and the opposition to repressive law enforcement agencies and practices [27–30].

Although violent protests and militant actions are the most visible forms of extremism, they are not required to be classified as extremist [22], as this would blur the distinction between extremism and other ideas such as fanaticism or terrorism (see [31]). Extremism therefore consists of two facets, a latent form in terms of attitudes and worldviews, and a manifest form that expresses itself in the respective behavior. In this study we focus on attitudes and make a clear distinction between extremist attitudes and violent behavior. Finally, we treat political extremism as a non-compensatory concept [32–34] that considers someone as extremist only when he or she holds strong left- or right-wing radical attitudes like socialism or social-Darwinism *and* anti-democratic attitudes at the same time. For illustration, Fig 1 provides a visual representation of the distinction between right-wing radical and extremist attitudes.

## Previous measurements and the need for new scales

So far, research on political extremism has mostly investigated apparent manifestations such as voting behavior. Unfortunately, this is often problematic, because behavior does not always equal attitudes. People may conceal their preferences owing to societal standards [35], or they may regard voting as a form of protest against the incumbent government. It may also be

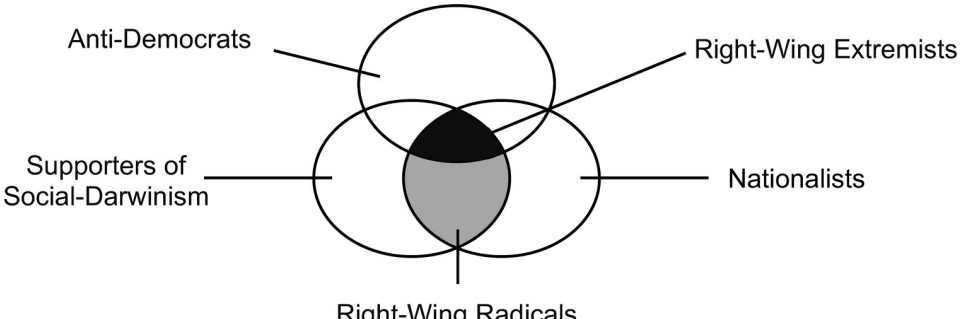

**Fig 1. Conceptualization of right-wing extremist attitudes.**

difficult to track the long-term development of political extremism if parties change their party platform, as shown in the situations of the PDS in Germany, the Socialistische Partij in the Netherlands, or elsewhere [36]. As a result, while the proportion of voters and radicals may overlap, this does not mean that they are the same thing.

Other popular measurements like ideological left-right self-placement face similar problems. Aside from the fact that people have a tendency to hide extreme opinions [5, 20], individuals vary in their understanding of what they consider as "left" and "right" [37]. Not only does this vary systematically across social groups, but also across time (e.g., [38]). Hence, both measurements fail to capture actual extremist attitudes in surveys as has been shown by Jungkunz ([9]).

In contrast to the huge corpus of research on right-wing extremist voting behavior, relatively few attempts at cross-national comparison of attitudes have been made. Scales are also not used in important international social surveys such as the European Social Survey, Eurobarometer, International Social Survey, or World Value Survey (see [25]). As a result, there appears to be a variable understanding of the measurement of extremist attitudes in different nations. A study for the Swiss National Science Foundation, for example, identifies three factors of right-wing extremism (authoritarianism and meritocracy, violence as an acceptable means, and distrust for the Swiss political system) which are supposed to be distinct from misanthropic measures such as sexism, Islamophobia, anti-Semitism, and xenophobia [39]. This is however contrary to numerous theoretical assumptions that assume that sexism, Islamophobia, and xenophobia are part of right-wing extremism, whereas violence acceptance is not a necessary element. Similarly, the European Union's sole research (SIREN) attempted to quantify extremist opinions by evaluating the most representative extreme right-wing party in each country [40]. Given that party agendas change between nations, this is a very problematic metric of right-wing extremism.

Other researchers developed the concept of Group-Focused Enmity (GMF), i.e. prejudice towards groups that are identified as "other" and are thus assigned an inferior social status [41]. The concept identifies sexism, racism, xenophobia, anti-Semitism, islamophobia and homophobia as subparts that share a common ideology. Nevertheless, it is not comparable to political extremism, as it lacks an anti-democratic element that is a necessary facet of any form of political extremism.

The biennial *Mitte-Studien* at the University of Leipzig in Germany produce the most thorough study on right-wing extremist attitudes [8, 42]. The studies developed a questionnaire for right-wing extremist attitudes that share features including chauvinism, xenophobia, anti-Semitism, social Darwinism, admiration of National Socialism, and support for autocracies. However, the studies' scale lacks psychometric validity, and conceptually, it merely considers anti-democratic attitudes as one aspect of right-wing extremism rather than as a required component. As a result, even if respondents only have low levels of support for dictatorship, they might nonetheless have high scores on the total scale.

As for left-wing radical and extremist attitudes there has been very little research in western nations as a whole. Neu offers a foundation for a list of items on left-wing extremism, but she analyzes the individual items of her "extremism scale" separately and does not build a full scale [5]. Furthermore, her work mixes radical and extremist attitudes with potential causes thereof at times. However, being one of the few researchers to examine both left- and right-wing extremism concurrently, she draws the important conclusion that there are indications of structural similarities between left- and right-wing extremists in their attitudinal structure. More recently, Schroeder and Deutz-Schroeder have provided an extensive assessment of prior studies on left-wing extremism in Germany [43]. In their own research, they generate a significant number of statements that are meant to map various aspects of left-wing

extremism. Despite the fact that the majority of the questions appear to be believable assertions, there has been no psychometric testing for the scale and it lacks items concerning authoritarian rule.

Jungkunz creates a battery of left-wing extremist or Marxian extremist attitudes over a twenty-year period by analyzing numerous pre-existing data sets [9, 44]. Unlike previous studies, he uses a measurement that considers someone to be an extremist only if left-wing radical attitudes such as support for socialism, nationalization, elite criticism, anti-capitalism, anti-imperialism, anti-Americanism, and GDR nostalgia coincide with anti-democratic attitudes. Unfortunately, due to data availability, psychometric testing cannot be applied to all items, and the analysis can only serve as a starting point.

Finally, Manzoni et al. developed scales for left-wing, right-wing and Islamistic extremism among Swiss adolescents [45]. Unfortunately, the authors explicitly combine left- and right-wing attitudes with the approval of the use violence against political opponents to form their overall scale, which is not in line with the above stated concept of political extremism that is supposed to be distinct from fanaticism and terrorism.

## Research design

We conducted a web survey in Germany (N = 2,117), Great Britain (N = 2,039), and the Netherlands (N = 2,045) between 21 June and 13 September 2022 using a recruited Bilendi & respondi online access-panel. The average response rate in the access panel is 40%. The target population were individuals of the public (age 18 to 69) in each country. Panel recruiting and membership is voluntary and based on a double-opt-in registration process. Participants provided informed consent in verbal form to participate in the study by clicking a button on the web interface before being able to start the survey. The survey was administered in each country's primary language and is representative in terms of sex, age (18 to 69 years), and education. The survey included an oversampling of Muslims for another reason. As a result, we weighted all models in *Mplus* [46]. Overall, about 50% of the respondents are female, about 57% have a degree that enables access to college and the mean age is 45.08 years (see Table A.1 in the S1 File). Quotas were realized through stratified random sampling. Because extremist attitudes are held by a small percentage of the population [9, 33], we chose countries where the far right and/or far left have recently had electoral successes. This should allow us to find a sizable number of respondents with extremist views and to test whether the scales work in different national contexts equally well.

We measure general extremist attitudes (GEX) with five items based on Likert-type response scales. In addition, we capture right-wing radicalism (RWR) with eight items and left-wing radicalism (LWR) with six items. The selection of items is based on substantive pre-testing and was narrowed down from an exhaustive list of items that we collected from previous research. We discuss this in greater detail in the S1 File A and B. Table 1 presents the question wording for all three scales. Descriptive summary statistics and information about missingness can be found Table A.1 and Figures A.1–A.14 in the S1 File.

After the various pretests the main survey included 20 to 25 items for each of the three scales. We then selected the items with the highest loadings in exploratory factor analyses. However, in cases where items achieved similar or very close factor loadings, we opted for items that added more conceptual breadth to the scale. Furthermore, we made sure to include items that include no country-specific information and which can be understood in the same way across countries.

We proceed by first testing the psychometric properties and construct validity of each sub-scale before we aggregate them to capture left- and right-wing extremists.

**Table 1. Question wording for political radicalism and extremism scales.**

| Item | Question wording |
|------|------------------|
| LWR1 | A decent living is only possible with socialism. |
| LWR2 | Capitalism is ruining the world. |
| LWR3 | Fascism shows the true face of capitalism. |
| LWR4 | The (nationality) foreign policy is racist. |
| LWR5 | The persecution of and spying on left-wing system critics by the state and police is increasing. |
| LWR6 | National states should be abolished. |
| RWR1 | We should have the courage to have a strong sense of national consciousness. |
| RWR2 | The (country) has become "too foreign" to a dangerous extent due to all the foreigners here. |
| RWR3 | Jews work more with evil tricks than others in order to get what they want. |
| RWR4 | Jews simply have something special and peculiar about them and do not really fit in with us. |
| RWR5 | Foreigners and asylum seekers are the ruin of (country). |
| RWR6 | Actually, (country) are inherently superior to other people. |
| RWR7 | We should make sure we keep (nationality) pure and prevent nations mixing. |
| RWR8 | The world would be a better place if people from other countries were more like (country). |
| GEX1 | It is better for government leaders to make decisions without consulting anyone. |
| GEX2 | People in government must enforce their authority even if it means violating the rights of some citizens. |
| GEX3 | Under some circumstances, a nondemocratic government can be preferable. |
| GEX4 | A concentration of power in one person guarantees order. |
| GEX5 | The government should close communication media that are critical. |

LWR: left-wing radicalism, RWR: right-wing radicalism, GEX: general extremism.

## Psychometric properties

In the following, we assess each scale's internal coherence, cross-national validity, and conceptual breadth. We first use confirmatory factor analysis (CFA) to determine whether the items of each scale load onto the dimension(s) they are attributed to. In order to assess cross-national validity, we follow-up with a measurement invariance test. Last but not least, we employ a model for graded ratings scales to generate information curves that show us how effectively these scales can distinguish between people on all levels of the latent political extremism traits.

We begin by determining whether each scale accurately captures the respective latent construct, i.e. political radicalism and extremism. To do so, we ran separate CFA for each scale in the pooled data and we examine first, how well the model fits the data, and second, the magnitude of the factor loadings. In all models, we take a reflective measurement approach (see further [47]). Table 2 gives an overview over model fit statistics and minimum and average loadings of each scale. Full results and results by country can be found in Appendix A in the S1 File. Overall, we find that all three scales have good to very good fit in RMSEA ($<0.08$), SRMR ($<0.08$) and CFI ($>0.95$) according to the cutoff-criteria by Hu and Bentler [48]. Finally, none of the scales holds particularly bad loadings, as the lowest standardized loading is 0.460 for LWR (see further [49]).

In a next step, we investigate whether the scales measure political radicalism and extremism in the same way across countries, i.e. measurement invariance. Ideally, we would expect that people from different countries with the same level of radicalism and extremism provide similar responses [50, 51]. If our political radicalism and extremism scales have the same meaning across countries, we could then compare e.g., mean differences in extremism scores or relationships between extremism scales and various predictors [52]. If, however, measurement invariance tests fail, the scales could not be used for cross-country comparisons, as we do not

**Table 2. Confirmatory factor analysis models on the pooled data.**

| | $\chi^2$ | df | p-value | RMSEA | SRMR | CFI | Min. Loading | Avg. Loading |
|---|---|---|---|---|---|---|---|---|
| Left-Wing Radicalism (LWR) | 95.699 | 8 | <0.001 | 0.044 | 0.022 | 0.978 | 0.460 | 0.611 |
| Right-Wing Radicalism (RWR) | 303.762 | 12 | <0.001 | 0.065 | 0.030 | 0.979 | 0.489 | 0.775 |
| General Extremism (GEX) | 14.122 | 5 | 0.015 | 0.018 | 0.008 | 0.998 | 0.619 | 0.675 |

Loadings represent standardized factor loadings.

know whether differences between countries are due to actual differences in scores or differences in response styles.

In general, a construct can achieve various degrees of measurement invariance: configural, metric, and scalar invariance. *Configural invariance* implies that the basic configuration in terms of the number of factors and the relationships between items and factor are the equivalent across countries. *Metric invariance* assumes that the magnitude of factor loadings is equivalent across countries. This type of invariance is at least necessary if researchers want to make cross-country comparisons between latent constructs and other predictors (e.g., regression coefficients), as it implies that a one unit change in the latent radicalism or extremism variable is equivalent across countries. Finally, *scalar invariance* assumes that item intercepts (in addition to factor loadings) are invariant. Since this is often considered a very strict assumption, it has become practice to allow the relaxation of some constraints in multi-group CFAs which is called *partial scalar invariance* [52]. However, there is no agreed rule upon an acceptable number of released constraints in partial scalar models (for a discussion see further [52]). Achieving (partial) scalar invariance enables researchers to also meaningfully compare latent means across countries. Hence, metric invariance is sufficient if researchers are interested in the relationship between political radicalism and extremism and predictors, as regression estimates are unbiased. However, if one wants to compare level of radicalism and extremism between countries, scalar invariance is necessary [50, 53].

In principle, we start by fitting a model with the same factor structure but different factor loadings and intercepts for each country (configural model). We then proceed by gradually forcing loadings (metric model) and finally intercepts (scalar models) to be the same across countries. The differences between models are usually assessed using $\chi^2$-tests. However, it is very difficult to achieve true invariance, especially for the stricter forms [54]. Furthermore, $\chi^2$-tests can easily become significant with larger sample sizes, hence picking-up minuscule differences between countries [55, 56]. We therefore use cutoff criteria for changes in RMSEA and CFI according to Chen ([55]) to determine whether the stricter model is still invariant as it decreases model fit not substantially. Based on these recommendations, changes in RMSEA between two models (e.g., configural and metric) of $\geq 0.015$ and changes in CFI of $\geq -0.01$ would indicate noninvariance.

Table 3 presents the summarized results from several multigroup CFA (MGCFA) that we used to test measurement invariance. Detailed results can be found in Tables A.11–A.13 in Appendix A in the S1 File. The results based on ΔRMSEA and ΔCFI show that all three scales achieve metric invariance meaning that they can be used to compare e.g., effect sizes of predictors for political radicalism and extremism across countries. Scalar invariance on the other hand is only achieved after relaxing at least the intercept of one item in one country (partial scalar invariance, see Tables A.11–A.13 in Appendix A in the S1 File for details). However, this is still sufficient to be able to compare means across countries [57].

**Table 3. Multigroup confirmatory factor analysis for measurement invariance across countries.**

| | LWR | | | | RWR | | | | GEX | | | |
|---|---|---|---|---|---|---|---|---|---|---|---|---|
| | RMSEA | ΔRMSEA | CFI | ΔCFI | RMSEA | ΔRMSEA | CFI | ΔCFI | RMSEA | ΔRMSEA | CFI | ΔCFI |
| Configural | 0.042 | | 0.981 | | 0.071 | | 0.976 | | 0.026 | | 0.995 | |
| Metric | 0.044 | 0.002 | 0.972 | 0.009 | 0.072 | 0.001 | 0.969 | 0.007 | 0.031 | 0.005 | 0.990 | 0.005 |
| Scalar | 0.060 | 0.016 | 0.936 | 0.036 | 0.085 | 0.013 | 0.950 | 0.019 | 0.075 | 0.044 | 0.923 | 0.077 |
| Scalar (partial) | 0.044 | 0.000 | 0.967 | 0.005 | 0.079 | 0.007 | 0.959 | 0.010 | 0.036 | 0.005 | 0.985 | 0.005 |

LWR: left-wing radicalism, RWR: right-wing radicalism, GEX: general extremism. Δshows differences in values between models: configural vs. metric, metric vs. scalar, and metric vs. scalar (partial).

In a final step, we test how much of the latent constructs (left- and right-wing radicalism and general extremism) our scales are able to capture using item response theory (IRT). IRT models can be very helpful to evaluate how good scales can discriminate respondents that hold high and low values of a particular latent concept. In our case, this means we can test whether our scales can distinguish radical and extremist respondents from non-radical and non-extremist ones.

In Fig 2 we display the information curves from graded rating scales models [58] in the pooled data for all three scales which contain the (standardized) range of the latent constructs on the x-axis and the quantity of information that the scale can capture on the y-axis. A scale works well across all points of the latent construct if the output shows a near to uniform distribution. The width of the distribution identifies the range of the latent concept that it discriminates well, with narrower distributions indicating less conceptual breadth. This could, for instance, indicate that a scale is able to differentiate low extremists from moderate extremists well, but it fails to distinguish moderate extremists from high extremists. The solid lines represents the information curves (left y-axis) where higher values indicate more information. The dashed line is the standard error which shows the amount of uncertainty in capturing the latent concepts (right y-axis). All three scales are standardized with a mean of zero and higher values indicate stronger radical and extremist attitudes.

Fig 2 shows that the left-wing radicalism scale has the broadest information curve from [-2,3]. Thus, it is able to capture respondents with weak and strong radical attitudes almost equally well. The scales for right-wing radicalism and general extremism have somewhat

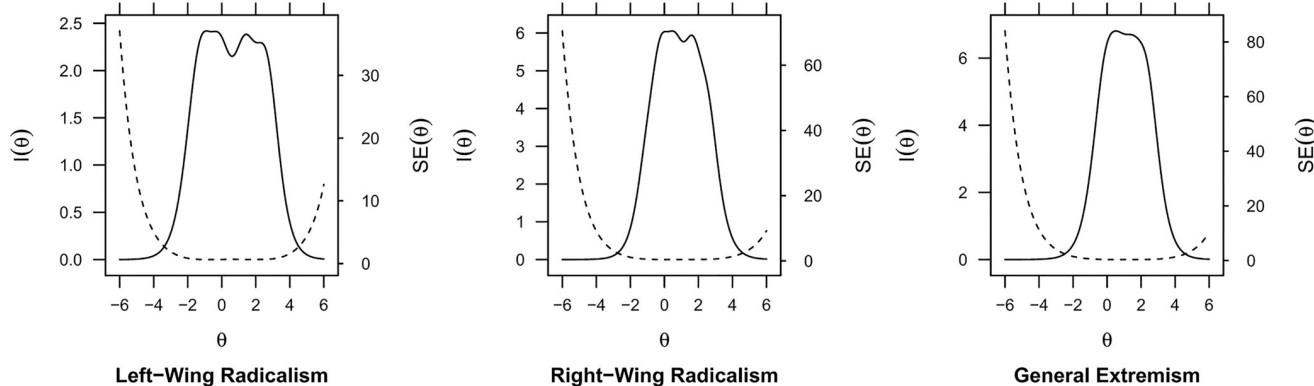

**Fig 2. Information curves for radicalism and extremism scales.** Based on pooled data. Information (solid lines) and SE curves (dashed lines) for radicalism and extremism scales. Higher values represent more radical and extremist attitudes.

narrower information curves in the [-1,3] range. However, we would argue that this is rather unproblematic in our case, as it implies that the scales are somewhat less able to distinguish weakly radical and extremist respondents from moderate radical and extremist respondents. In turn, they are still very able to distinguish respondents with moderate radical and extremist attitudes from those with strong radical and extremist attitudes.

## Construct validity

Another major indicator of scale assessment concerns a scale's external validity. We therefore evaluate the construct validity of the three scales by testing how well they are connected to known correlates. If the scales work well, they should at least show moderate correlations. We therefore chose four concepts that are closely related to political radicalism and extremism: authoritarian personality traits, conspiracy beliefs, political detachment, and political violence justification. We used a twelve item battery by Oesterreich ([15]) for authoritarian personality traits ($\alpha = 0.767$), the five-item scale by Bruder et al. ([59]) for conspiracy beliefs ($\alpha = 0.878$), seven items for political detachment ($\alpha = 0.913$), and two separate batteries for general and specific political violence justification (general: $\alpha = 0.836$, [60]; specific: $\alpha = 0.925$). In addition, we used one item on a six-point scale to capture respondents' general support for the idea of democracy and the left-right self-placement scale ranging from zero (very left) to eleven (very right).

Table 4 shows the Pearson correlations between radicalism and extremism scales and related constructs in the pooled data (separate models by country in Tables A.14–A.16 in the S1 File indicate similar findings). As we would expect, left- and right-wing radicalism and general extremism are all moderately and positively correlated with conspiracy beliefs, political detachment, and both political violence justification indices. In most cases, correlations are in the range between 0.30 and 0.60, whereas conspiracy beliefs and political detachment are somewhat lower for general extremism and political violence justification somewhat higher for general extremism. Authoritarian personality traits are, however, only associated with right-wing radicalism and general extremism, but not left-wing radicalism. Again, this is in line with previous research, as left-wing *radicals* do not necessarily possess a rigid worldview as left-wing *extremists* do (see further [12, 61, 62]). Furthermore, we find moderately to strongly negative correlations with the idea of democracy, indicating that radicals and extremists dislike the idea of democracy. Finally, the left-right scale shows that the scales work well in the expected directions, as we find a negative correlation with left-wing radicalism and a positive one with right-wing radicalism. We also find that general extremism is more correlated with

**Table 4. Correlations between radical and extremist attitudes and related constructs.**

|  | LWR | RWR | GEX |
|---|---|---|---|
| Authoritarianism personality traits | -0.010 | 0.278 | 0.217 |
| Conspiracy beliefs | 0.381 | 0.319 | 0.191 |
| Political detachment | 0.356 | 0.270 | 0.139 |
| Political violence justification: general | 0.317 | 0.510 | 0.563 |
| Political violence justification: specific | 0.333 | 0.481 | 0.577 |
| Idea of democracy | -0.269 | -0.351 | -0.423 |
| Left-right scale | -0.206 | 0.391 | 0.239 |

Based on pooled data. LWR: left-wing radicalism, RWR: right-wing radicalism, GEX: general extremism. Higher values on the left-right self-placement scale represent a more conservative, right-wing oriented placement.

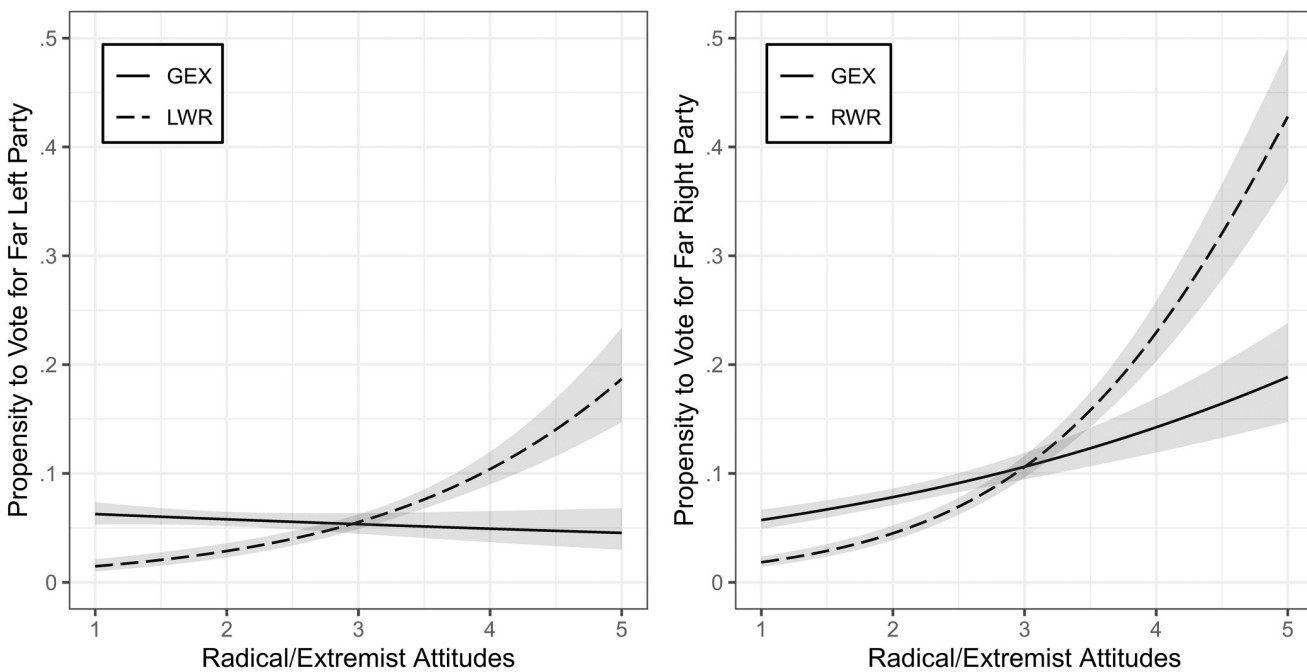

**Fig 3. Propensity to vote for far left and far right parties by radicalism and extremism scales.** Based on logistic regressions with 95% confidence intervals. LWR: left-wing radicalism, RWR: right-wing radicalism, GEX: general extremism. Higher values represent more radical and extremist attitudes.

considering oneself as right-wing or conservative but considerably less than compared to the right-wing radicalism scale.

Finally, radicalism and extremism scales should also be able to predict vote choice for such parties. Our sample includes multiple parties that can be considered as left- or right-wing radical. To increase the sample size, we combined voting for the Left Party in Germany and the Socialistische Partij in the Netherlands to far left party vote, and voting for the AfD in Germany, the UKIP in Great Britain, and PVV and FvD in the Netherlands to far right party vote. While these parties are generally regarded as left- or right-wing radical, their degree of political extremism is clearly very heterogeneous. For instance, the Left Party in Germany moved closer to the political center since reunification and holds the minister-presidency in the state of Thuringia since 2014 [63]—similar to the development of the Socialistische Partij in the Netherlands [64]. The AfD on the other hand moved further to the right and parts of the party are now officially labelled as extremist. In fact, the German Federal Office for the Protection of the Constitution recently considered parts of the party as *Verdachtsfall* (subject of extended investigation to verify a suspicion, for more information, see https://www.verfassungsschutz.de/SharedDocs/pressemitteilungen/EN/2022/press-release-2022-1-afd-1.html). Thus, we would expect that the radicalism scales work well to predict far left and far right party vote, but that the results should be more mixed in the case of general extremism.

Fig 3 shows the propensity to vote for a far left party (left panel) and far right party (right panel) by radicalism (dashed lines) and extremism scales (solid line). As expected, higher values on the left- and right-wing radicalism scales indicate a greater likelihood to vote for far left respectively far right parties. For instance, the propensity to vote for a far left party is near zero if someone holds very low left-wing radical attitudes (value of one) but increases to 19 percent for respondents with very high values (five points). For right-wing radicalism, the effect is even stronger and the propensity to vote for far right parties increases from two percent to 43

percent from very low to very strong levels. The general extremism scale works very differently though. As we can see in the left panel, the curve is almost flat, indicating no relationship between general extremism and voting for far left parties (from six percent to five percent). In the panel on the right, we see, however, that there is quite a substantial effect on far right party vote, as the probability increases from six percent to 19 percent.

Taken together, we conclude that our scales show strong signs of external validity. In fact, we argue that the non-effect of the general extremism scale on far left party vote is actually a good indicator, as the parties in our sample can be classified as left-wing radical but not as politically extremist. Thus, our scales are able to discriminate well between radicalism and actual extremism.

## Aggregating left- and right-wing extremism scales

Last but not least, we want to highlight some guidelines for aggregating or combining the scales. As we explained in the beginning of the paper, political extremism is a non-compensatory concept. Thus, someone has to hold strong right- or left-wing radical attitudes *and* anti-democratic attitudes at the same time to be considered as left- or right-wing extremist (see Fig 1). Non-compensatory concepts are challenging for operationalization and scholar often use simple mean or additive indices to combine the two dimensions of political extremism (see further [33, 34]). While methodologically more refined, this also applies to second-order factors in CFA that first form separate factors for radicalism and anti-democracy and then an overarching factor as a combination of the two factors.

To adequately combine left-wing and right-wing radicalism with general extremism, we propose a multiplication method that assures that respondents that are characterized as left- or right-wing extremists hold high values on both dimensions at the same time. Although the proportions of respondents considered holding extremist attitudes are comparable to the ones presented in the main text, they are probably not identical respondents, as the correlation between both aggregation methods is about 0.9 for left-wing and right-wing extremism (see Figures A.12 and A.13 in the S1 File). We therefore suggest to first create two separate indices (either based on mean indices or factor scores) of the radicalism and general extremism items. These should then be recoded so that the lowest value is zero. In our case here, we rescale both indices to a range from zero to four. In the second step, we multiply both scales to form an overall measure of either left- or right-wing extremism.

The advantage of this method is that it penalizes overall scores if a respondent has a low value on any dimension. Additionally, because all individual items and sub-dimensions are coded to a range from zero to four, it is guaranteed that respondents cannot have extremely low values (i.e. zero) on one dimension because doing so would provide a zero value for the total index. Strong views on both dimensions lead to notably high values on the total score, which in turn, derives from holding such attitudes. Thus, despite taking into consideration all the characteristics, our technique reflects (albeit not entirely) the non-compensatory nature of left- and right-wing extremism (see further [65]).

An alternative method would be to take the minimum value across both dimensions (see further [33, 34, 65]). Unfortunately, this method's main drawback is that it entirely ignores any data from dimensions other than the one with the lowest value. One may thus have extremely high scores on left-wing radicalism and very low scores on anti-democracy and be categorized as having very low total scores. Nevertheless, we provide the results from this aggregation method as additional robustness check in Figure A.10 in the S1 File. The findings show slightly lower shares for left- and right-wing extremists which is likely due to the above mentioned

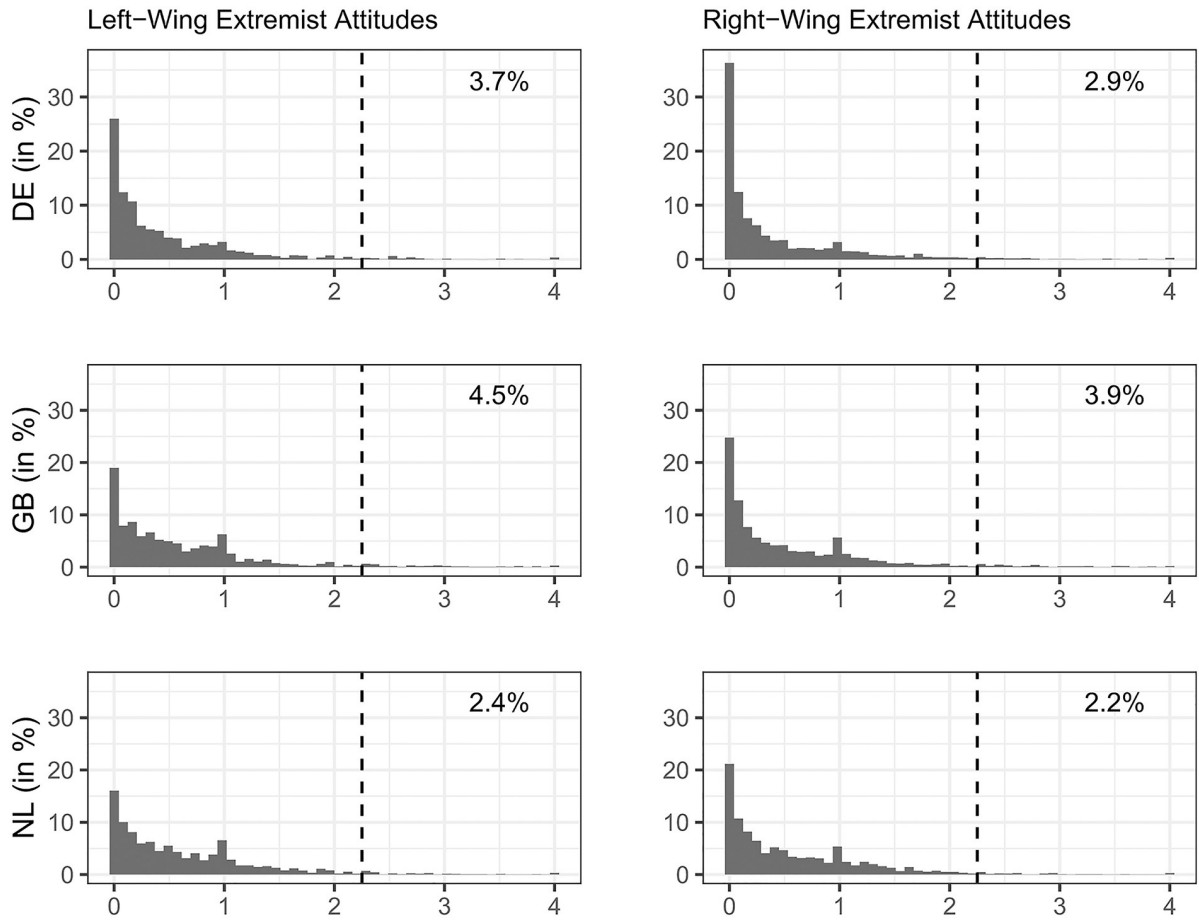

**Fig 4. Distributions of left- and right-wing extremist attitudes by country.** Shown are histograms with percentage of respondents by degree of left- and right-wing extremist attitudes by country. Higher values indicate stronger extremist attitudes. Numbers in the plot area refer to the share of respondents that on average at least "agreed" with all items.

restrictions. Finally, we also provide the distributions based on a non-compensatory additive mean index in Figure A.11 in the S1 File.

To give an impression how prevalent left-and right-wing extremism are in societies, we show in Fig 4 the distributions and shares of respondents with extremist attitudes in Germany, Great Britain and the Netherlands. In sum, we find that the distributions are very skewed, as most people hold either no or only low levels of extremist attitudes. Thus, the shares of respondents who could be characterized as having a left-wing extremist worldview is quite low with 3.7 percent in Germany, 4.5 percent in Great Britain and 2.4 percent in the Netherlands. Similarly, 2.9 percent in Germany can be identified as having a right-wing extremist worldview, along with 3.9 percent in Great Britain and 2.2 percent in the Netherlands. Thus, *extremist* political attitudes can be found only among a small fraction of the population. It does, however, not imply that radical political attitudes are equally rare. For instance, the Leipzig Authoritarianism Study reports that about 17.0 percent of Germans hold manifested xenophobic attitudes in 2022 [66]. In our data, combining the xenophobia related items RWR2 and RWR5 from Table 1 results in a slightly lower number of 13.4 percent of Germans who can be characterized as xenophobic.

## Discussion and conclusion

In this paper we developed three new scales for left- and right-wing radical and general extremist attitudes. Our results show that all three scales have high internal and external validity and work equally well across three Western European countries. Given the lack of validated scales for attitudes towards political radicalism and extremism, our work provides a major step towards the investigation of the causes and consequences of radical and extremist attitudes. As of now, scholars mainly investigated voting for far left or far right parties or used ad hoc measures like the left-right self-placement scale to assess radical and anti-democratic tendencies in society. Unfortunately, both options are subject to measurement error, either because we cannot distinguish protest voters from true believers, or because people have a different understanding of what is considered left and right. The advantage of our scales is that they can capture latent radical and extremist attitudes among respondents who have not yet openly committed to support radical and extremist parties. Thus, identifying such individuals allows us to investigate the radicalization process of individuals at a much earlier stage.

A limitation of our scales (so far) pertains to their geographical scope. Although we developed the scales in three different countries, they are all located in Western Europe. While we assume that they can be used in other countries of Western Europe and North America as well, future research could investigate their applicability in Eastern Europe and other regions of the world. Although we particularly included items that can be understood without national context, cultural differences might impair universal applicability. Furthermore, while our scales are able to discriminate between radicalism and extremism, they are only theoretically confined from other concepts like fanaticism or terrorism. Future research could therefore investigate the relationship between these different concepts empirically.

Finally, we have to admit that our proposed items—like all surveys with sensitive questions—can underlie some sort of social desirability bias. Experimental strategies that aim to avoid such bias and are able to identify implicit extremist attitudes should therefore complement traditional survey research (see further [4]). This would provide particularly interesting insights, for instance by investigating the relationship between implicit and explicit extremist attitudes. Thus, we could gain more insights into when and how citizens begin to openly state their extremist attitudes. Ultimately, this would allow us to further elucidate the process of radicalization (see further [21]).

## Supporting information

**S1 File. Supporting information.**
(PDF)

## Author Contributions

**Conceptualization:** Sebastian Jungkunz, Marc Helbling.

**Data curation:** Sebastian Jungkunz, Nina Osenbrügge.

**Formal analysis:** Sebastian Jungkunz, Nina Osenbrügge.

**Funding acquisition:** Sebastian Jungkunz, Marc Helbling.

**Investigation:** Sebastian Jungkunz, Nina Osenbrügge.

**Methodology:** Sebastian Jungkunz, Nina Osenbrügge.

**Project administration:** Sebastian Jungkunz, Marc Helbling.

**Resources:** Sebastian Jungkunz, Marc Helbling.

**Software:** Sebastian Jungkunz, Nina Osenbrügge.

**Supervision:** Sebastian Jungkunz, Marc Helbling.

**Validation:** Sebastian Jungkunz, Marc Helbling, Nina Osenbrügge.

**Visualization:** Sebastian Jungkunz, Nina Osenbrügge.

**Writing – original draft:** Sebastian Jungkunz, Marc Helbling.

**Writing – review & editing:** Sebastian Jungkunz, Marc Helbling, Nina Osenbrügge.

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
