## [Decision Letter · Decision Letter 0]

4 Jan 2024

PONE-D-23-35231Measuring political radicalism and extremism in surveys: three new scalesPLOS ONE

Dear Dr. Jungkunz,

Thank you for submitting your manuscript to PLOS ONE. After careful consideration, we feel that it has merit but does not fully meet PLOS ONE’s publication criteria as it currently stands. Therefore, we invite you to submit a revised version of the manuscript that addresses the points raised during the review process.

**I found two referees with expertise in political psychology for your manuscript**.

**They confirmed that the manuscript was well written and did not contain any major problems. They also recognized the importance of developing a scale to measure extremist attitudes using a large survey sample. I agree with them**.

**However, both reviewers asked you to address some points. Both reviewers pointed out the inadequacy of the reporting of results and the need to mention limitations. See the individual comments for other points. Please revise your manuscript according to their comments with point-by-point responses.**

We look forward to receiving your revised manuscript.

Kind regards,

Yutaka Horita

Academic Editor

PLOS ONE

Journal Requirements:

Deutsche Forschungsgemeinschaft (DFG) - Project number 438614532

Reviewers' comments:

Reviewer's Responses to Questions

**Comments to the Author**

1. Is the manuscript technically sound, and do the data support the conclusions?

Reviewer #1: Yes

Reviewer #2: Yes

2. Has the statistical analysis been performed appropriately and rigorously? 

Reviewer #1: Yes

Reviewer #2: Yes

3. Have the authors made all data underlying the findings in their manuscript fully available?

Reviewer #1: Yes

Reviewer #2: No

4. Is the manuscript presented in an intelligible fashion and written in standard English?

Reviewer #1: Yes

Reviewer #2: Yes

5. Review Comments to the Author

Reviewer #1: Review of Measuring political radicalism and extremism in surveys: three new scales

This is a very good, insightful and neat study. I strongly recommend it to be published subject to minor revisions. Below are some suggestions.

“right- and left-wing radicalism” … “understand people who take drastic or far-reaching positions on issues such as nationalism or anti-imperialism” (p.2): To me that is not a particularly precise definition. First I would try to define it without the “or” and leave out the “drastic” (what is drastic?). Second, I wonder whether one could refine what the authors understand by “far-reaching”.

The authors mention a few other concepts that should be differentiated from extremism such as fanaticism, terrorism. Generally, the focus is on convergent validity and less on discriminant validity albeit the authors write that “our scales are able to discriminate well between radicalism and actual extremism”. (p.14). To me it’s a limitation of the current paper that the discussion and empirical differentiation from other concepts gets not spotlight. Hence, I would add this to the conclusion as a venue for future research and point the way where to go and to look.

“Finally, we treat political extremism as a non-compensatory concept [24–26] that considers someone as extremist only when he or she holds strong left- or right-wing radical attitudes like socialism or social-Darwinism and anti-democratic attitudes at the same time.”: The authors’ definition implies that someone CANNOT be considered an extremist if they only hold strong left- or right-wing radical attitudes without the anti-democratic component. This has quite far-reaching consequences (if everyone would really follow their advice). While the authors cite others I would like them to add a paragraph that more extensively explains this decision in the present paper. What are the pros and cons of adding the anti-democratic component to the definition and why do you do it?

Could there be extremists that hold radical beliefs but do not hold anti-democratic attitudes? For example, could there be an individual that holds extreme environmental views and engages in non-violent protest but does not necessarily oppose democratic principles. I guess following the authors this person would not be an extremist. A discussion of interesting edge cases would be extremely helpful.

And, how do environmental or religious extremism fit into the author's definition? Someone who is radically anti-abortion but pro-democratic ist not an extremist I guess. Could these other forms of extremism be included? (religious extremism is mentioned in the appendix page 2).

In measurement we can make a fundamental distinction between formative and reflective indicators/measurement. It would be helpful if the authors could point out which approach they are following with which method, e.g., see Coltman et al. (2008).

“The survey was administered in each country’s primary language and is representative in terms of sex, age, and education” (p.7): A comparison of the survey data to census statistics would be great (for all three countries). Right now the reader has no way to check that statement. In addition, the authors should outline how representativity was reached (quota sampling? weighting?). In principle, quota sampling could be stated as a limitation in the conclusion for transparency reasons.

There is no discussion of response rates and not analysis of missing data. Both would be good to have in the appendix. The naniar R package is quite useful to visualize missingness in the dataset.

“Finally, none of the scales holds particularly bad loadings, as the lowest standardized loading is 0.460 for LWR.” (p.22): What are acceptable cutoffs here?

Table 1: Right now there is no discussion of the single items that go into the scales. However, I imagine that a lot of discussions have happened prior to the selection of those items. Ideally, the authors would add a bit more background (even if only in the appendix) on the selection of those items.

Table B1: Why not show the shares of different educational levels?

Figure 2: If you are allowed to print color you could combine the plots into one (with three different colors) which would facilitate the comparison that you describe in the text.

Table 4: This is pooled across all 3 countries. However, it probably also makes sense to show this data unpooled for the three different countries in the appendix.

Aggregation & Multiplicative index (see below): I am not completely sold on this idea and got a bit carried away. On the one hand we somehow need to combine the two dimensions (radicalism + general extremism) to follow the author's definition. I tried to do a very simple simulation in R with the code attached for the multiplicative index (which might be wrong). Underlying the same value of the multiplicative index there may still be variation in the underlying dimensions. Hence, I would recommend the authors to alert readers to the potential of such variation. For some values of the multiplicative index the two sub dimensions may diverge quite strongly. This could matter once we relate the index to other dimensions. In addition, it would be nice to see such distributions for the empirical data that was actually collected by the authors in the appendix. The theoretical range of the index may not reflect the empirical one.

Future research: The authors mention social desirability as one challenge. However, I imagine that there are other venues of research that might spring from this paper (i.e., what would the authors like to do in the future) that should be discussed in the conclusion. In part, these should be linked to the limitations the present study shares with others, e.g., restricted to few countries, few time points, potentially biased sample representativity of the sample with regard to political orientations etc.

Finally, I found a data availability statement but I wonder about the embargo period. I would expect the authors to publish raw data & code files to reproduce all analyses and graphs upon publication. The raw data can be a subset of the larger survey I guess.

library(tidyverse)

library(plotly)

leftwing_extremism <- runif(1500, min = 0, max = 4)

general_extremism <- runif(1500, min = 0, max = 4)

combined_score <- leftwing_extremism*general_extremism

data <- bind_cols(leftwing_extremism = leftwing_extremism,

general_extremism = general_extremism,

combined_score = combined_score)

# Choose different ranges for index here to better compare variation

data <- data %>% filter(combined_score > 6 & combined_score < 6.5)

plot_ly(x=data$leftwing_extremism,

y=data$general_extremism,

z=data$combined_score,

type="scatter3d",

mode="markers",

text = ~paste('

leftwing_extremism: ', round(data$leftwing_extremism,1),

'

general_extremism: ', round(data$general_extremism,1),

'

combined_score: ', round(data$combined_score,1))) %>%

layout(scene = list(aspectmode = "manual", aspectratio = list(x=1, y=1, z=1),

xaxis = list(title = 'leftwing extremism',

range= c(0,4)),

yaxis = list(title = 'general extremism',

range= c(0,4)),

zaxis = list(title = 'combined score',

range = c(0,max(combined_score)))

))

Reviewer #2: The present study developed three novel scales, applicable across Western European nations, to examine right-wing, left-wing, and general extremist attitudes (RWR, LWR, GEX). Utilizing data from Germany, Great Britain, and the Netherlands, the researchers examined the validity of these scales by evaluating the psychometric properties of the derived indices, including their internal consistency and capacity to differentiate among strongly extremist individuals. Furthermore, the researchers investigated the relationships among the scales and various constructs associated with extremist attitudes to establish external and construct validity. The findings substantiate the high validity of these three scales, which are applicable across Western European nations.

Overall, the manuscript is well-written, and the results are clearly presented. I have a few comments regarding relatively minor issues, which are explained below.

1. (p. 2, lines 13–23) The tone of the sentences in this section could be modulated to present a more balanced perspective. It is important to note that several established standardized measures, such as the right-wing authoritarianism scale, attempt to assess related constructs, even though their primary focus may not be right- or left-wing extremism.

2. The authors repeatedly refer to the Appendix, but the precise location of the relevant information within the Appendix remains unclear.

3. (pp. 9–10, lines 208-241) Clarification is needed regarding the distinction between a scalar model and a partial scalar model. Additionally, is the context of ΔRMSEA and ΔCFI significant regarding the difference between these two types of models?

4. (p. 11, lines 261–269) I want to make sure I understand this portion correctly. Is the claim based on the right-skewed peak in the information curves and the fact that SE curves do not have high values on the right side in Figure 2?

5. Descriptive summary statistics are provided in the supplementary material. However, the authors should include these details in the main text because information such as participant age and gender is vital to the readers.

6. The study should present the correlations between RWR, LWR, and GEX. This would enable a discussion of the claim that GEX is non-compensatory, based on various findings in the study.

7. In Table 1, please consider adding the mean and standard deviation for each item within each scale.

8. The study should acknowledge its limitations. Every research study has constraints and acknowledging them helps temper the claims made. Please list two to four limitations.

6. PLOS authors have the option to publish the peer review history of their article (what does this mean?). If published, this will include your full peer review and any attached files.

Reviewer #1: No

Reviewer #2: No

---

## [Author Response · Author response to Decision Letter 0]

15 Jan 2024

We uploaded a separate document in which we detail all changes as response to the reviewers comments.

---

## [Decision Letter · Decision Letter 1]

4 Mar 2024

Measuring political radicalism and extremism in surveys: three new scales

PONE-D-23-35231R1

Dear Dr. Jungkunz,

We’re pleased to inform you that your manuscript has been judged scientifically suitable for publication and will be formally accepted for publication once it meets all outstanding technical requirements.

Kind regards,

Yutaka Horita

Academic Editor

PLOS ONE

Additional Editor Comments (optional):

Reviewers' comments:

Reviewer's Responses to Questions

**Comments to the Author**

1. If the authors have adequately addressed your comments raised in a previous round of review and you feel that this manuscript is now acceptable for publication, you may indicate that here to bypass the “Comments to the Author” section, enter your conflict of interest statement in the “Confidential to Editor” section, and submit your "Accept" recommendation.

Reviewer #1: All comments have been addressed

Reviewer #2: (No Response)

2. Is the manuscript technically sound, and do the data support the conclusions?

Reviewer #1: Yes

Reviewer #2: (No Response)

3. Has the statistical analysis been performed appropriately and rigorously? 

Reviewer #1: Yes

Reviewer #2: (No Response)

4. Have the authors made all data underlying the findings in their manuscript fully available?

Reviewer #1: Yes

Reviewer #2: (No Response)

5. Is the manuscript presented in an intelligible fashion and written in standard English?

Reviewer #1: Yes

Reviewer #2: (No Response)

6. Review Comments to the Author

Reviewer #1: (No Response)

Reviewer #2: I thank the authors for taking my comments into consideration. In general, I think the paper has improved through the changes made. I think that this manuscript is publishable in PLOS ONE.

7. PLOS authors have the option to publish the peer review history of their article (what does this mean?). If published, this will include your full peer review and any attached files.

Reviewer #1: No

Reviewer #2: No
